# The Feeding Behaviour Habits of Growing-Finishing Pigs and Its Effects on Growth Performance and Carcass Quality: A Review

**DOI:** 10.3390/ani12091128

**Published:** 2022-04-28

**Authors:** Marta Fornós, Santos Sanz-Fernández, Encarnación Jiménez-Moreno, Domingo Carrión, Josep Gasa, Vicente Rodríguez-Estévez

**Affiliations:** 1Cargill Animal Nutrition, 50170 Mequinenza, Spain; fornos.marta@gmail.com (M.F.); encarnacion_jimenez@cargill.com (E.J.-M.); domingo_carrion@cargill.com (D.C.); 2Department of Animal Production, Universidad de Córdoba, 14071 Córdoba, Spain; v22safes@uco.es; 3Animal Nutrition and Welfare Service, Department of Animal and Food Sciences, Universitat Autònoma de Barcelona, 08193 Bellaterra, Spain; josep.gasa@uab.cat

**Keywords:** growing-finishing pig, feeding behaviour, carcass, growth performance

## Abstract

**Simple Summary:**

The study of feeding behaviour habits (FBHs) of growing-finishing pigs is of interest due to its influence on growth performance and carcass quality. The present review collates the available scientific data regarding the internal and external factors affecting the FBHs and its influence on growth performance and carcass quality. Factors explored were age, sex, breed, space allowance, feeder design, feed form, diet composition, and environmental conditions. The reviewed data indicate that the factors explored affect the FBHs of growing-finishing pigs. Moreover, meal size and feeding rate were the two FBHs most related with performance, being positively correlated with average daily feed intake, growth rate, and final body weight, but with no clear effect on feed efficiency, whereas the few studies regarding the influence of FBHs on carcass traits indicate a positive correlation between meal size and feeding rate with backfat thickness. Therefore, the available data provide evidence that modifying FBHs may improve the performance of grow-ing-finishing pigs, but not necessarily feed efficiency.

**Abstract:**

Based on the available data of feeding behaviour habits (FBHs), this work aimed to discuss which type of pig, according to its FBHs, performs better and is more efficient. As pigs grow, average daily feed intake, meal size, and feeding rate increase, whereas small variations or even decreases in time spent eating and daily feeder visits have been reported. Moreover, the sex, breed, space allowance, feeder design, feed form, diet composition, and environmental conditions modify FBHs. On the other hand, the literature indicates the existence of four types of pigs: pigs that eat their daily feed intake in many short meals (nibblers) or in few large meals (meal eaters) combined with eating fast (faster eaters) or slow (slow eaters). The available scientific literature about ad libitum fed pigs suggests that pigs eating faster with bigger meals eat more, gain more weight, and are fatter than pigs eating less, slower, and with smaller meals. However, the feeding rate and the meal size do not influence feed efficiency. In conclusion, studies comparing growing-finishing pigs with similar feed intake, but different feeding rate and meal size are needed to better understand the influence of FBHs on feed efficiency.

## 1. Introduction 

Feed cost represents approximately 65% of the cost production of a pig kg deadweight [1]. Therefore, the search for strategies to improve the utilisation rate of nutrients during the growing-finishing period is of permanent interest [2,3,4]. One of the important factors influencing the performance and carcass quality of growing-finishing pigs is feeding behaviour habits (FBHs) [5], which can be described not only by average daily feed intake (ADFI), but also by other criteria such as the daily number of feeder visits, the daily time spent eating, the feed consumed per feeder visit or the rhythm of ingesta, among others. These can be registered and calculated thanks to the availability in the market of automatic feeding systems [6]. 

It is known that the ADFI is directly related to energy and nutrient intake [7]; however, FBHs influence the digestion and absorption of feed nutrients [2,8]. Meal size is one of the factors that influences the digestibility of nutrients [9]. In fact, de Haer et al. [10] reported that meal size and feeding rate influence the growth performance of growing-finishing pigs, with pigs eating small meals and slower being leaner and with a lower average daily gain (ADG), with poor influence of the number of meals and the time spent eating with performance. In addition, Carcò et al. [5] concluded that the feeding rate is the most correlated FBH parameter with growth performance being positively related with ADG and final body weight (BW). Furthermore, few studies have evaluated the influence of FBHs on carcass quality traits [5,10,11]. 

The first aim of the present review was to collate and compare data showing the effect of internal (age, sex, and breed) and external factors (group size and feeder space allowance, feeder design, feed distribution and feed form, diet composition and environmental conditions) on the FBHs of growing-finishing pigs. The second aim was to collate and compare the published data regarding the influence of FBHs on the growth performance and carcass quality traits of growing-finishing pigs. The implications of FBHs as a strategy to improve performance and carcass quality are summarised.

## 2. A Description of the Feeding Behaviour Habits of Growing-Finishing Pigs 

Table 1 and Figure 1 include the different published criteria used to describe the FBHs of growing-finishing pigs and its interrelation, respectively. The FBH parameters were average daily feed intake (ADFI, total feed consumed per pig and day), feeder visits per day (TV, number of feeder visits per pig and day), meals per day (TM, number of meals per pig and day), time spent eating (TD, total time spent eating per pig and day), visit size (VS, feed consumed per feeder visit), meal size (MS, feed consumed per meal), and feeding rate (FR, feed intake per minute spent eating). 

A determinate number of feeder visits conducted consecutively within a period by the same pig are often clustered into one meal [6,10,11,12,13,14,15,16]. However, the period selected between feeder visits conducted consecutively by the same pig to determine a meal varies from one minute [15] to 28.3 min [16] between studies. Therefore, when comparing FBH parameters such as MS between studies, it is important to know the criteria used to define one meal. We suggest that the standardisation of a criterion to define a meal is of interest. 

## 3. Internal Factors That Influence Feeding Behaviour Habits of Growing-Finishing Pigs

### 3.1. Age 

A summary of the effect of age on the FBHs of growing-finishing pigs is shown in Table 2. As pigs grow, the ADFI increases; however, the magnitude of the ADFI increase is variable among studies. Labroue et al. [11] and Andretta et al. [15] reported an increase in the ADFI of around 60% in pigs of similar BW, from 35 to 95–100 kg BW and from 30 to 100 kg BW, respectively; whereas Carcò et al. [5] reported a smaller quadratic increase in the ADFI in pigs from 47 to 145 kg BW and Hyun et al. [16] obtained an increase in the ADFI of 23% in pigs from 27 to 82 kg BW. On the other hand, pigs eat their ADFI from frequent feeder visits in weaned pigs to few and larger feeder visits in sows together with an increase in the FR [17,18]. The changes in the TV and VS may be due to larger stomach size as pigs grow. In fact, stomach size increases from 30 mL to 3.5 L from birth to a finishing pig [19]. Therefore, we hypothesize that 20 kg BW pigs ingesta could be limited by their stomach capacity and as a consequence, carry out a higher number of small feeder visits to achieve the desired ADFI. For instance, as growing-finishing pigs grow, ADFI, VS, MS, and FR increase, whereas small variations or even decreases in the TV, TM, and TD have been reported [5,11,15,16,20]. However, a large variability in the percentage of increase or decrease in all FBHs exists between studies. In terms of TV or TM, Labroue et al. [11] reported an increase in TV of 28% in pigs from 40 to 60 kg BW and a reduction of 11% in pigs from 60 to 90 kg BW; whereas Hyun et al. [16] and Gonyou and Lou [20] obtained a reduction of 17% in the TM and of 24% in the TV, respectively, in pigs of similar BW. In addition, Andretta et al. [15] and Carcò et al. [5] reported small variations in terms of TM and TV as pigs grew, respectively. On the other hand, reductions from five to 45% in the TD [11,15,16,20] and increases from 45 to 123% in the VS or MS [11,15,16] together with increases from 22 to 133% in the FR as pigs grow have been reported [11,15,16,20]. 

### 3.2. Sex 

The contradictory results regarding the effect of sex on the FBHs shown in several studies could be due to the different level of competition access to the feeder [11,15,16,21,22,23,24,25]. No differences between sex in terms of the FBHs of growing-finishing pigs were found in the meta-analysis of Averós et al. [21]. Similarly, Hyun et al. [16] only found differences between sexes in terms of TM, being higher for castrated males than for entire males and females; whereas Andretta et al. [15] reported no differences in terms of TM between castrated males and females. On the other hand, Cross et al. [22] observed that females spent an average of 6.2 min per day less in the feeder than castrated males, a result in line with the findings of Brown-Brandl et al. [25]. Moreover, Pichler et al. [23] observed bigger and longer meals for growing-finishing entire males than for females with no other FBHs showing differences between sex. In contrast, Young and Lawrence [24] observed a tendency for smaller and shorter feeder visits in entire males than females. In addition, Andretta et al. [15] reported a 19.23% smaller MS for females compared to castrated males. Furthermore, Labroue et al. [11] reported lower MS, ADFI, and TD in entire males than in castrated males with no significant differences in terms of TM, TV, and FR between both groups. Furthermore, Andretta et al. [15] indicated that females had a 6.6% lower FR than castrated males (39.9 vs. 42.7 g/min, females and castrated males, respectively). 

### 3.3. Breed

Breed modifies the FBHs of growing-finishing pigs [26,27,28,29]. Fernández et al. [26] classified Large White and Pietrain pigs as nibbler pigs due to more frequent and smaller feeder visits per day than Duroc and Landrace pigs. These results are in keeping with the findings of Labroue et al. [27], who reported more frequent smaller feeder visits for Large White than for Landrace pigs. Likewise, Baumung et al. [28] observed that Large White pigs ate their ADFI in more TV, with less TD and lower FR, whereas Landrace pigs tended to eat their ADFI in fewer and larger feeder visits. In addition, Quiniou et al. [29] concluded that Pietrain pigs could be characterised by eating their ADFI in more frequent, smaller meals than Meishan pigs, with Large White pigs in an intermediate position. On the other hand, Landrace and Large White pigs were classified as fast eater pigs due to the fact that they spent less TD with higher FR than Duroc and Pietrain pigs [26]. In agreement with those results, Labroue et al. [27] reported smaller differences in terms of FR with an average of 39.9 g/min for Large White and 41.5 g/min for Landrace pigs. In fact, Fernández et al. [26] suggested that each breed could be described as follows: Duroc pigs as meal and slow eaters, Landrace pigs as meal and fast eaters, Large White pigs as nibblers and fast eaters, and Pietrain pigs as nibblers and slow eaters. 

Despite the inconsistencies among studies of the impact of age, sex, and breed on the FBHs, all of them indicate that the three factors influence FBH. Although different intervals of BW were evaluated in the cited studies, it was found that as pigs grow, ADFI, MS, and FR increase, while decreases or small variations in TD, TV, and TM occur. The results concerning the sex effect on FBHs are confusing, suggesting that the external conditions such as housing conditions or internal factors such as age or breed used could modify FBHs. In fact, most of the authors observed different FBHs when comparing different breeds. Therefore, when comparing the FBH results of different scientific data sources, these factors must be considered. 

## 4. External Factors That Influence Feeding Behaviour Habits of Growing-Finishing Pigs

### 4.1. Group Size and Feeder Space Allowance

The EU Directive 2008/120/EC [30] determines the minimum stocking density for growing-finishing pigs at different BWs, which is an important factor, as it is demonstrated that it affects the stress levels of growing-finishing pigs [31]. In addition, later studies have observed that increasing group size in growing-finishing pigs in an adequate pen floor space and feeder ratio does not impact their welfare and growth performance [32]. These results suggest that an important factor is feeder access competency. In fact, it has been observed that individually housed pigs eat their ADFI in smaller, more frequent meals, spending more TD on account of a lower FR than group-housed pigs [12,33]. Moreover, when increasing the group size from two to 12 growing pigs per pen (from 27 to 48 kg BW) with the same stocking density of 0.9 m^2^/pig and with a single-space feeder, pigs reduced the TD and increased the FR with lower ADFI and ADG with no effect on the feed conversion ratio (FCR) [34]. When increasing the group size from five to 20 pigs per pen in 34 kg BW pigs for 29 days keeping the same stocking density of 1.06 m^2^/pig with a single-space feeder, pigs ate their DFI in fewer and larger feeder visits with higher FR with no impact on performance results (no differences in ADFI, ADG, and FCR) [35]. In finishing pigs, the increase from two to 12 pigs in group size increased the TD, MS, and FR and reduced the TV with no effect on ADFI, ADG, or FCR [36]. Therefore, these results suggest that growing-finishing pigs may modify their FBHs due to the feeder-space restricted situation rather than due to the increase in group size. In fact, Averós et al. [21] predicted that pigs fed under feeder space-restricted conditions increase their FR, make shorter feeder visits, and reduce the TD, results in agreement with Gonyou and Brumm [37]. In fact, Nielsen et al. [38] suggested that the FR may be used as an indicator of social constraint. Therefore, not only is pen floor space important, but it is also important to have the correct feeder ratio. In fact, an insufficient ratio of feeders in group-housed growing-finishing pigs may limit the nutritional requirements of the pigs. However, what does an adequate feeder ratio mean? Linear feeder space is defined as “the linear cm of feeder available per pig within a pen” (total feeder length per pen/total pigs per pen). PIC [39] recommends a minimum between 4.7 and 5.0 cm per pig for dry feeders and between 2.9 and 3.1 cm for wet–dry feeders in pigs from 27 kg BW to target BW to minimize feed waste without decreasing the ADFI of pigs. In fact, Smit et al. [40] observed that 3.4 cm of linear feeder space per pig in wet–dry feeders was enough as they obtained the same growth and final BW with lower ADFI than pigs with one more extra feeder, suggesting that the extra feeder allowed pigs to waste feed. Moreover, Morrison et al. [41] compared growing entire males pigs housed in deep-litter (pen of 200 pigs with 1 m^2^/pig and 8.3 pigs/feeding space) vs. pigs housed in conventional system (pen of 45 pigs with 0.70 m^2^/pig and 8.5 pigs/feeding space) from 20 to 22 weeks of age and observed that pigs housed in deep-litter spent less TD, with fewer and larger feeder visits, with a lower frequency of social interactions around the feeder compared to pigs in conventional treatment, concluding that the competency between pigs in the conventional system may be responsible for the shorter and more frequent feeder visits and that pigs are able to modify their FBHs in order to maintain performance under limitations in feeder space. In this sense, Rodríguez-Estévez et al. [42] found that free range pigs modified their foraging group size depending on the grazed resource, with 5.0 animals/group when pigs were grazing in an open pasture versus 5.8 when they were eating acorns under an oak crown because they were conditioned by the crown space to avoid competition when foraging, sharing a mean grazing surface to forage acorns of 8.9 m^2^/pig.

On the other hand, growing-finishing pigs showed two peaks of feed intake throughout the day (one in the morning and another in the afternoon) [15,16,33], which has also been observed in free range finishing pigs grazing natural resources [43]. During these two peaks, which are accentuated under heat stress conditions [22], the competition access to the feeder increases. In fact, increasing the group size from 10 to 30 pigs increased the feeder occupancy rates due to increased feeding activity during the night and at midday [44], whereas increasing group size from 18 to 22 with an extra feeder allowed pigs to eat according to their preferent diurnal pattern instead of eating at other moments of the day [40]. Moreover, the hierarchy within a pen also influences the FBHs with fewer and larger visits for the high-ranking pigs than the low-ranking pigs [45]. Therefore, under feeder space restrictions, the hierarchy may distinctly modify FBH. These results highlight the importance of analysing the FBH at an individual level. In fact, the authors of the present review have presented a new approach [non-published study] to detect the maintenance of the FBHs at an individual level and broadly, the results indicate that most pigs maintain their FBHs throughout the growing-finishing period, except for ADFI, which is the most difficult FBH to predict. 

### 4.2. Automatic Feeding Systems Used to Record Feeding Behaviour Habits 

Different types of automatic feeding systems exist in the market to record the FBH of group-housed growing-finishing pigs. In Table 3, a summary of the automatic feeding systems used and the FBH measured in previous studies is presented. In these systems, pigs are individually identified with a data-carrying transponder with a unique code per pig detected by the reader system installed in the trough [46]. Most of the systems record the start and end time, the duration and the amount of feed intake of each feeder visit, and the pig BW can be registered by the installation of a load cell; from these data, the different FBH parameters can be calculated. 

**Table 3 animals-12-01128-t003:** Summary of the automatic feeding systems used and of the feeding behaviour habits measured in previous studies.

Feeding Behaviour Parameter	IVOG-Station (Figure 2)	Compident Pig-MLP (Figure 3)	ACEMA 48 (Figure 4)	F.I.R.E., Hunday Electronics	Similar System to the Used in Hyun et al. [16]	Recording System in a Commercial Trough (See Figure 5)
ADFI ^1^	[4,8,10,12,26,47,48]	[5,49]	[11]	[16]	[34,36]	
TV ^2^	[4,8,10,12,26,47,48]	[5,49]		[16]	[34,36]	
TM ^3^	[8,10,12,26]		[11]	[16]		
TD ^4^	[4,8,10,12,26,47,48]	[5,49]	[11]	[16]	[34,36]	[25]
MS ^5^	[8,10,12,26]		[11]	[16]		
VS ^6^	[4,8,10,12,26,47,48]	[5,49]		[16]	[34,36]	
FR ^7^	[4,8,10,12,26,47,48]	[5,49]	[11]	[16]	[34,36]	

^1^ ADFI (average daily feed intake). ^2^ TV (number of feeder visits per pig and day). ^3^ TM (number of meals per pig and day according to each paper methodology; where a meal is: the successive feeder visits within five minutes [10]; the successive feeder visits within two minutes [11]. Carcò et al. [5] analysed the daily number of feeder visits. ^4^ TD (total minutes spent eating per pig and day). ^5^ MS (feed consumed per meal). ^6^ VS (feed consumed per feeder visit). ^7^ FR (feed intake per minute spent eating).

**Figure 2 animals-12-01128-f002:**
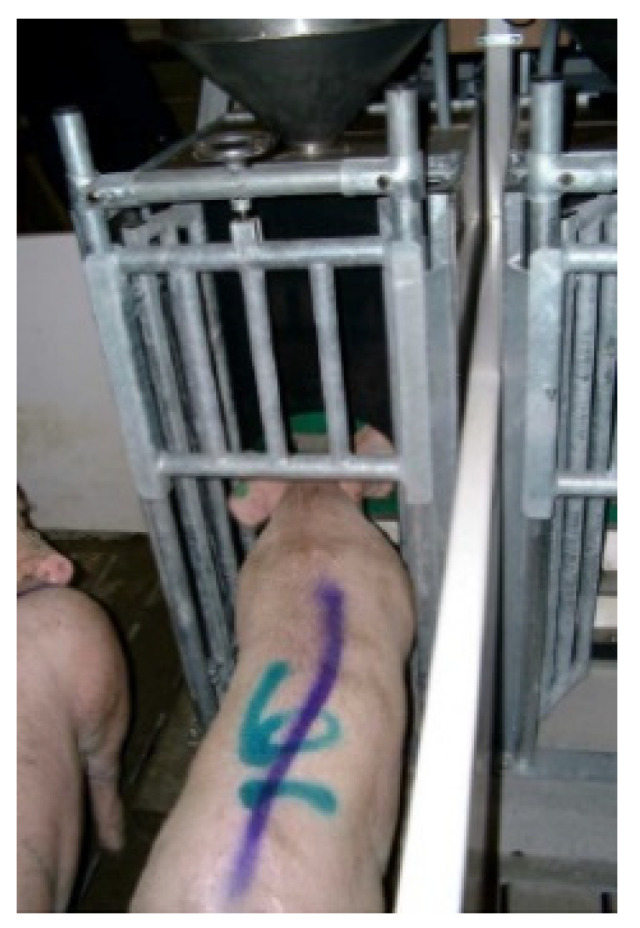
IVOG—A station for individual feed intake recording in group housing (Instentec B.V., Marknesse, the Netherlands) used in the studies of De Haer and Merks, [12], De Haer et al. [10], De Haer and de Vries, [8], Georgsson and Svendsen, [47,48], Rauw et al. [4], and Fernández et al. [26] (Source: [www.insentec.eu], accessed on 5 April 2022).

**Figure 3 animals-12-01128-f003:**
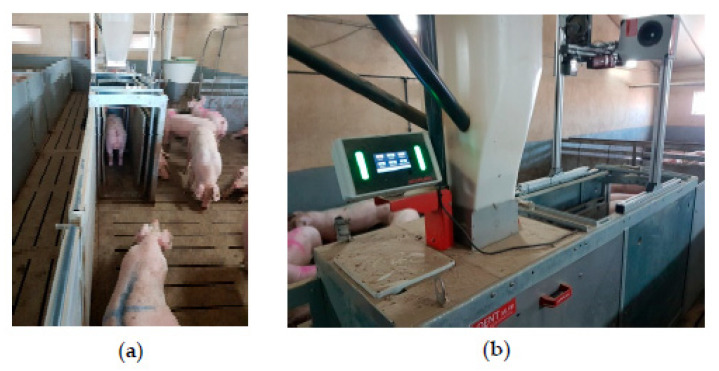
Compident MLP (Schauer Agrotonic GmbH, Austria) used in the study of Garrido-Izard et al. [49]. (**a**) Weighing scale. (**b**) Feeding station used during the experiment (Source: [49]).

**Figure 4 animals-12-01128-f004:**
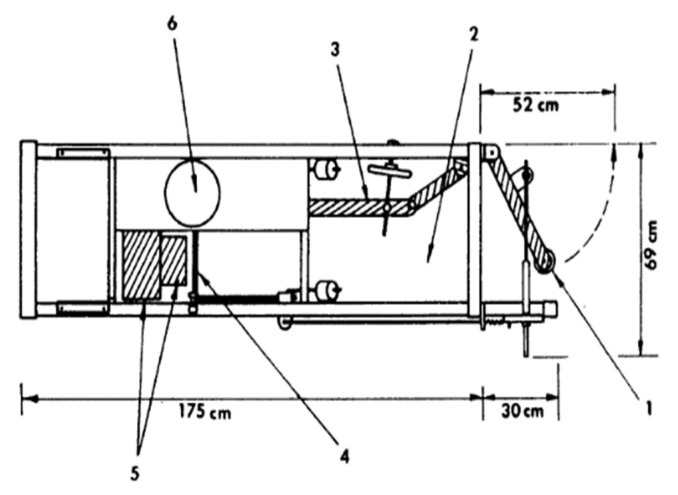
Electronic feeding station referred to as ACEMA “48” used in the study of Labroue et al. [11]. (1) Access door to the feeder. (2) Access corridor to the trough. (3) Adjustable side. (4) Trough door. (5) Feed hopper. (6) Mechanism to fill up the trough (Source: [11]).

**Figure 5 animals-12-01128-f005:**
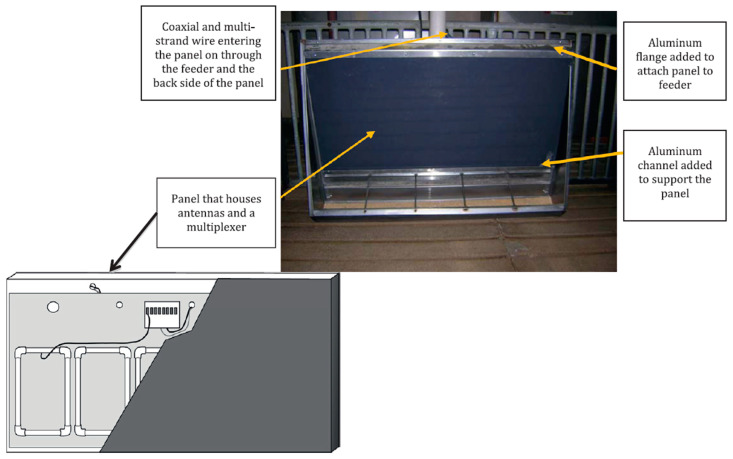
Schema of the panel and a photo of the panel after installation (Source: [25]).

One of the available automatic feeding systems is the IVOG-Station (Individual feed intake recording in group housing, Instentec B.V., Marknesse, the Netherlands; Figure 2). This system consists of a dry-single space feeder placed on load cells with an adjustable fence that provides head and neck protection for the pig in front of the feeder. This system has been used in the studies of De Haer and Merks, [12], De Haer et al. [10], De Haer and de Vries, [8], Georgsson and Svendsen, [47,48], Rauw et al. [4] and Fernández et al. [26]. 

Another type of automatic feeding system is the Compident Pig-MLP (Schauer Agrotonic, Austria; Figure 3), which can feed growing-finishing pigs ad libitum and ration up to four different feeds at the same time and was used in the study of Carcò et al. [5] with lateral barriers to avoid competition among the pigs during the feeder visit together with a gate placed in front of the trough that permits only one pig inside the feeder. In the study of Garrido-Izard et al. [49], the Compident MLP (Schauer Agrotonic GmbH, Austria) was also used and equipped with an individual animal scale with lateral barriers to determine individual animal weight from 35 to 120 kg BW by measuring the weight of the front and back parts of the pig. 

Labroue et al. [11] used a system referred to as “ACEMA 48” (Figure 4). This system consists of a trough, which allowed them to weigh the feed and a gate to avoid the entrance of more than one pig into the trough at the same time. Feed is weighed before and after each feeder visit and if the amount of feed after the visit of a pig is below 400 g, the hopper is refilled up to 1200 g. 

Hyun et al. [16] used recording equipment (F.I.R.E., Hunday Electronics, Newcastle-upon-Tyne, UK) consisting of a trough connected to a load cell equipped with a full-length protective crate to prevent the entrance of more than one pig at any time. Hyun and Ellis [34,36] used a similar feed intake recording system with a crate in front of the trough (Osborne Industries, Osborne, KS). On the other hand, Brown-Brandl et al. [25] developed a system to record the TD per pig in a commercial trough by a radio frequency identification system in growing-finishing pigs (Figure 5). 

It is known that the type of automatic feeding system used influences FBH of growing-finishing pigs [6,48]. Therefore, due to the existence or not of lateral barriers to protect the head and neck while the pig is eating, or due to the presence or not of a gate to prevent the access of more than one pig to the feeder, the FBHs differ. In fact, the model of the meta-analysis of Averós et al. [21] predicted that the use of protection barriers within individual feeders increased the TD and reduced the TV, FR, and FCR compared to when using feeders without protection barriers. Moreover, Bruininx et al. [50], comparing weaning pigs allotted in the IVOG feeding station versus pigs allotted in commercial single-space dry feeders for 34 days, obtained higher ADFI during the first 13 days for the pigs reared in the IVOG system, but during the remaining 21 days and overall, the ADG and the FCR did not differ between systems. In growing-finishing pigs, a higher ADFI and poorer FCR were obtained in pigs allotted in IVOG stations compared to conventional feeders [47], whereas similar ADG but lower ADFI and FCR were reported in growing- [34] and finishing pigs [36] fed by electronic feeders compared to those pigs fed by conventional feeders. The reasons for the lower ADFI or improved FCR in pigs fed by electronic feeders compared to conventional feeders may be a consequence of the lower feed waste due to the design of the feeder or because only one pig can access the trough of the automatic feeding systems at any one time, reducing the competency in the feeder if it is compared to conventional feeders. 

### 4.3. Feed Form and Feed Distribution

Growing-finishing pigs can be fed with different feed forms (mash or pelleted feed), with different water level availability in the feeder (dry feeders or wet–dry feeders) and by different feed distribution systems (ad libitum or restricted). Therefore, in this subsection, a review of the available scientific data regarding the effect of those factors on FBHs and performance of growing-finishing pigs is presented (Table 4). MacDonald and Gonyou [51] reported that growing-pigs (35–45 kg BW pigs) and finishing-pigs (90–100 kg BW) spent more time eating when feed was in dry mash than in dry pellet form. On average, pelleted fed pigs spent 11.5% less time eating than mash fed pigs. Those results are in agreement with Li et al. [52], who reported a 23.5% and a 37.1% reduction in the TD in growing and finishing pigs, respectively, with pigs fed with pellets compared to pigs fed with mash; furthermore, the pigs fed with pelleted feed had a higher FR and a lower feeder occupancy rate. These results are in concordance with Laitat et al. [53], who observed that weaned pigs needed more time to achieve the same ADFI when feeding a mash diet than a pelleted diet due to lower FR. 

MacDonald and Gonyou [51] and Li et al. [52] analysed the combined effect of feed form (mash vs. pellet) and water availability (dry vs. wet–dry feeders) in growing-finishing pigs. In both growing (20 to 60 kg BW) and finishing (60 to 100 kg BW) pigs, Li et al. [52] observed an interactive effect of feed form and water availability with the dry-mash fed pigs spending a longer time eating due to their lower FR than any other treatment. These results are consistent with the previous findings of MacDonald and Gonyou [51]. In addition, Gonyou and Lou [20] also observed that growing-finishing pigs fed ad libitum by wet-dry feeders spent 17% less time eating than pigs fed by dry feeders, suggesting that growing-finishing pigs prefer wet–dry to dry feeders [40]; furthermore, pigs fed by wet–dry feeders had higher ADFI and ADG and pigs were less lean. In the study of Li et al. [52], the effect of feed form and water availability on performance was analysed in growing and in finishing pigs. In both phases, water availability did not influence FCR, the most efficient pigs being those fed a pelleted diet. Additionally, FBHs of growing-finishing pigs differed when the same feed was offered: dry or dry feed diluted with water (88.6 vs. 27.8% dry matter, dry and dry-feed diluted, respectively) twice per day; growing-finishing pigs fed with dry feed diluted with water spent around 50% less time than pigs fed with dry feed with no differences in terms of performance [54]. 

On the other hand, the meta-analysis of Averós et al. [21] reported that pigs fed restrictively ate in longer feeder visits and were more active, perhaps because the pigs visited the feeder to check whether there was feed available, than pigs fed ad libitum. On extensive farms, in which pigs have access to restricted feed together with ad libitum access to fodder and grass, the feeding behaviour of pigs depends on a large number of factors such as the dietary supplementation, grazing management, and grass quality, among others [55]. 

### 4.4. Diet Composition

Several studies have evaluated the effect of diet composition on the FBHs of growing-finishing pigs. The main factor that modifies the ADFI of a pig is the energy content of the diet; a pig fed with a low energy diet eats more feed per day compared to a pig fed with a high energy diet in order to achieve the required daily energy [40]. In fact, the dilution of the energy concentration of the diet can be carried out by increasing the dietary fibre level, which may be used as a strategy to reduce stereotypic behaviour and to enhance welfare by its satiety effect after a meal by reducing feed motivation [56,57]. In fact, pigs fed with a low nutrient density spent longer eating per day and per feeder visit compared to pigs fed with a higher nutrient density diet [23]. In addition, Quemeneur et al. [58] concluded that the inclusion of fibre (a mix of wheat, soy, and sugar beet pulp fibres) decreased meal frequency, increased MS, whereas the supplementation of aleurone decreased the TM with no effect on MS. On the other hand, lysine content in the diet reduced the number and increased the length and size of feeder visits [16]. Carcò et al. [59] observed that pigs increased ADFI and tended to increase the FR with reduced amino acid content in the diet to achieve nutritional requirements. Furthermore, the flavour and the palatability of feed may stimulate the appetite of pigs. In fact, the inclusion of flavouring additives such as dextrose increases the ADFI of pigs, although there are discrepancies about this fact in the literature [7]. On the other hand, Iberian finishing pigs under extensive conditions depending on natural resources without compound feed remain active, foraging acorns and grass an average of 369 min per day, which is approximately 60% of winter daylight hours; this kind of slow eating would be very dependent on the natural diet [60].

### 4.5. Environmental Conditions

The effect of high temperature on ADFI, pig activity, and performance has been widely studied [13,21,61,62]. The meta-analysis of Renaudeau et al. [63] shows that the reduction in ADFI and ADG under high temperature is higher in heavier than in lighter growing-finishing pigs (Figure 6).

However, few studies have evaluated the effect of environmental conditions on the FBHs of growing-finishing pigs (Table 5). In growing pigs (from 21 to 30 kg BW), Collin et al. [14] reported a reduction of 30% in ADFI, 32% in MS, and 27% in TD with a negative impact on BW gain (−37%) after thirteen consecutive days at 33 °C compared to the control group reared at 23 °C. In heavier pigs (62 kg BW), a decrease of 24% in ADFI, 21% in TV, and 28% in TD were observed when the temperature was increased from 19 to 29 °C for three or four consecutive days at 19, 22, 25, 27, or 29 °C [13]. In fact, Cross et al. [22] observed a reduction of approximately four minutes in TD when growing-finishing pigs were under heat stress conditions. The reduction in ADFI under heat stress is probably a strategy to reduce body heat production [64], which comes from maintenance, physical activity, and feed intake [61]. 

Moreover, the feed intake schedule changes under different environmental conditions. Under hot conditions, pigs reduce their physical activity [61] and spend more time lying and less time eating [65]. Cross et al. [22] observed that under thermoneutral conditions, most feeder activities were carried out from 6:00 to 17:59 h, while when pigs were suffering heat stress, a peak feeding activity occurred between 6:00 and 08:59 h, a reduction during midday, and another peak of feeder activity between 18:00 and 20:59 h in all breeds and genders studied. 

The reviewed scientific data regarding the effect of external factors on the FBHs of growing-finishing pigs highlights the importance of the knowledge of each of the factors explored as all of them impact on the FBHs. In intensive conditions, pigs are allotted in groups in pens that can differ in terms of size, number, and type of feeders or stocking density, among others. The reviewed data indicate that growing-finishing pigs are able to adapt their FBHs to achieve the desired ADFI to maintain growth. Therefore, depending on housing conditions, pigs change their FBHs. On the other hand, feed form and feed distribution influence the FBHs; pigs fed in dry mash spend more time eating than pigs fed in dry pelleted feed due to lower FR, whereas when water is available in the feeder, their ADFI and FR increase, but with no influence on FCR. These results indicate that the feeder occupancy rates are higher when pigs are fed in mash, suggesting that the stocking density recommended could depend on the feed form offered. Continuing with parameters related with diet, its composition is of high importance. It is widely known that ADFI depends mainly on diet energy density, with a higher ADFI in pigs fed with low-density diets than pigs fed with high-density diets. However, the type of fibre used or the amino acid content can also modify the FBHs of growing-finishing pigs. Finally, the magnitude of the impact of environmental conditions on ADFI was higher in older than in younger pigs, also distinctly affecting the FBHs depending on the age. 

## 5. Feeding Behaviour Typologies 

In this section, the correlations between the FBH parameters of growing-finishing pigs reported in the available scientific data are presented [4,11,12,16,24,26,49] (Table 6). De Haer and Merks [12] and Labroue et al. [27] distinguished two types of pigs by their number and size of meals: “*nibbler*” pigs (many short meals every day) and “*meal eater*” pigs (a few long meals every day). In fact, strong and negative correlations between MS and TV have been reported, indicating the existence of pigs eating many short meals and pigs eating a few large meals [11,12,16,24,26,49]. Moreover, Fernández et al. [26] also found a strong and positive correlation between VS and the duration of the feeder visits in all of the breeds studied (Duroc, Landrace, Large White, Pietrain r ≥ 0.87; *p* < 0.05), suggesting no differences in terms of FR between *nibbler* and *meal eater* pigs. Moreover, the authors also classified pigs by their rhythm of ingesta, distinguishing “*fast eaters”* and *“slow eaters”.* This classification is supported by the strong and negative correlation reported by the available scientific data between FR and TD, indicating that pigs with a higher FR spend less time eating [4,11,12,16,24,26,49] whereas low correlations have been reported between TV and MS with TD and FR [4,11,12,16,24,26,49]. Therefore, the correlations of the reviewed scientific data suggest and support the four feeding behaviour typologies suggested by Fernández et al. [26] in growing-finishing pigs based on the number and size of the daily feeder visits (*nibbler* and *meal eater* pigs) and on the rhythm of ingesta (*fast* and *slow* eater pigs): *nibbler*-*fast eater, nibbler*-*slow eater, meal*-*fast eater*, and *meal*-*slow eater* pig.

## 6. The Relation between Feeding Behaviour Habits and Growth Performance 

In this section, the correlations between the FBH parameters and performance results of growing-finishing pigs are presented [4,5,10,11,12,16,24,26,49] (Table 7 and Table 8). Broadly, the correlations reported between the FBHs and growth performance are moderate with a maximum of 0.59 observed between TD and ADFI.

It is well-known that ADFI is directly related with energy and nutrient intake [7] whereas the size and frequency of meals affect the digestibility of nutrients [2,9]. It follows that the use of feed energy and nutrients depends on different metabolic mechanisms, which may be modified by FBHs such as meal frequency [9,66]. In humans, Schwarz et al. [67] and Toschke et al. [68] showed that, besides calorie intake, TM and MS are additional factors that affect BW and body composition whereas in pigs, MS and FR are the two FBHs most strongly and positively related with ADFI, ADG, and BW; however, the former have little effect on FCR [4,5,10,11,12,16,24,26,49].

Labroue et al.’s [27] results suggested that breeding to increase appetite would lead to *fast meal* eater pigs instead of *nibbler* pigs and concluded that MS and FR are the two FBH parameters most related with performance and are correlated with ADG. In agreement with these results, Carcò et al. [5] found that FR was the most highly correlated FBH with ADFI, final BW, and ADG; however, it was not correlated with gain to feed ratio and they suggested that the manipulation of FR would affect feed intake and as a consequence, growth performance. Likewise, Andretta et al. [15] found a negative correlation between MS and FR with gain to feed ratio, suggesting that MS and FR negatively influence nutrient utilisation, probably as a consequence of its effects on the passage rate or digestive enzyme activity [8,10]. However, only four studies have been found regarding the influence of MS and FR on feed efficiency and all have reported low correlations [5,11,16,49]. The only FBH parameter with significant influence on FCR was the TD with a positive correlation [5,11,16,49], which suggests that pigs spending a shorter time eating have better FCR. Nevertheless, these results are in contrast to pigs grazing on natural resources because most of the energy intake (54.1%) is to cover maintenance requirements [60]. In summary, the correlations reported by the reviewed authors suggest that increases in FR are associated with higher ADFI, higher growth rates, and less TD; in addition, increases in MS are associated with higher ADFI and higher growth rates. However, these increases in FR and MS did not show any influence on feed efficiency.

Controversial correlations have been reported between TV and performance [4,5,10,11,12,16,24,26,49]. In fact, de Haer and Merks [12] reported a positive correlation of TV with ADFI and ADG whereas Labroue et al. [11], Hyun et al. [16], Rauw et al. [4], and Fernández et al. [26] reported negative correlations, with neither of the cited studies showing an influence on FCR. Moreover, Schulze et al. [69] concluded that TV is independent from growth performance in boars. However, various authors have evaluated the effect of feeding frequency (feeding pigs at certain intervals of time during the day) on the performance of growing-finishing pigs with contradictory results. In the 70s, Allee et al. [70] reported that 22 kg BW pigs fed ad libitum were less efficient than pigs fed a single daily meal (2 h/24 h). A later study with heavier pigs (from 25–35 to 100 kg BW) also concluded that the more efficient pigs individually housed had fewer meals per day and shorter TD with higher MS [10]. In addition, Le Naou et al. [66] observed that 30 kg BW pigs allotted in individual cages and fed with the same amount of feed twice per day improved their ADG by 6.4% and their FCR by 4% compared to pigs fed 12 times per day, results which are in agreement with Liu et al. [71]. These results could be explained because pigs with fewer meals per day may reduce their maintenance requirements [72]. The energy requirements of pigs are divided into two fractions: energy needed for production and energy needed for maintenance. Energy for maintenance is defined as *“the level of feeding at which the requirements for energy are just met to ensure the continuity of vital processes so that there is no net gain or loss of energy and nutrients in tissue or animal products”* [73]. However, energy requirements for maintenance change depend on the physical activity of the pig. In fact, compared to resting, when a sow is standing, she almost doubles her body heat production [74] and McDonald et al. [75] reported that body heat production rate increases by 95% above the resting level when a 40 kg BW pig is standing. Van Milgen et al. [73] observed that body heat production due to activity represented between eight and 13% of the metabolizable energy intake in growing pigs. Therefore, it could be hypothesised that more meals per day would increase the energy requirements for maintenance and therefore penalize performance. In addition, pigs fed once or twice are generally less sensitive to the excitement associated with the distribution of feed than animals receiving multiple small meals, wasting less energy [76]. However, Schneider et al. [77], studying the effect of restricted feeding frequency from six to two meals per day with a similar amount of feed provided in both treatments (68 and 114 kg BW pigs allotted in pens of 10 pigs) observed a positive effect of the number of meals, with an increase in ADG and an improvement in FCR. Similarly, Colpoys et al. [78] obtained lower ADG and ADFI in growing gilts fed twice per day than fed ad libitum with no effect on FCR. These results are partially in agreement with those reported by Jia et al. [2], who concluded that feeding the same daily amount of feed once, twice, or five times a day modified digestion processes and performance. In fact, ADG, together with the apparent total tract digestibility of protein and fat, improved with five feeding times per day compared to feeding only once per day, however, those pigs obtained poorer FCR. Therefore, the reviewed studies indicate that, in restricted fed pigs, the frequency of feeding modifies performance. Thus, it could be hypothesised that a change in feeding frequency for pigs under a restricted feeding regime could modify MS and FR. Furthermore, in pigs fed ad libitum, this hypothesis could explain the low, contradictory correlations reported between TV and performance results, while MS and FR have been strongly correlated with ADFI and ADG but not with FCR [4,5,10,11,12,16,24,26,49].

In summary, most of the papers reviewed showed a positive influence of TD, MS, and FR on ADFI, whereas only MS and FR were mostly positively related with ADG; the influence of TV on ADFI and ADG was not clear, together with low and contradictory correlations between the FBHs and FCR.

## 7. The Relation between Feeding Behaviour Habits and Carcass Quality Traits

Despite the big economic interest in achieving specific carcass quality traits, few studies have evaluated the influence of FBHs on carcass quality traits (Table 9). The three found studies reported strong and positive correlations between ADFI, MS, and FR with backfat thickness whereas one of the two found studies showed strong and negative influences between ADFI, MS, and FR with lean percentage [5,10,11]. These results suggest that pigs eating large and faster meals may be fatter than pigs eating small and slower meals. In the same direction, Rauw et al. [4], studying growing-finishing pigs (Duroc barrows) allotted in group and fed ad libitum, observed that the pigs that ate faster, ate more, and spent less time eating and had higher fat deposition values. Similarly, Kavlak and Uimari [79] reported positive correlations between FR and backfat thickness and Stote et al. [80] and Toschke et al. [68] concluded that large energy intake meals led to higher adipose tissue deposition than eating smaller meals in humans. In addition, Carcò et al. [5] observed a high influence of FR on carcass quality traits in grouped housed pigs. In fact, it was observed that pigs eating faster had higher carcass weight, higher proportion of fat in the carcass, and lower proportions of carcass lean cuts than pigs eating slower (12.6 vs. 38.2 g/min). However, Colpoys et al. [78] did not find any correlation between FR, ADFI, ADG, protein or fat deposition, and lean estimated by X-ray tomography; their study was conducted with a small number of gilts fed ad libitum or twice a day. On the other hand, low correlations have been reported between TV and TD with carcass quality traits [5,10,11]. In terms of feed efficiency, the literature indicates that pigs eating more and faster grow faster and are fatter, but with no effect on feed efficiency [4,5,10]. In summary, despite finding only a few studies, the results regarding the correlation between FBHs and carcass quality suggest that pigs that eat more, with higher MS, and eat faster, may have thicker backfat thickness and lower lean percentage values.

## 8. Conclusions

First, since several definitions of a meal can be found in the literature, it is recommended to standardise the criteria or use the parameter feeder visit instead of the meal concept.

Second, it was confirmed that the feeding behaviour of growing-finishing pigs is influenced by internal and external factors. Therefore, when analysing the feeding behaviour of growing-finishing pigs, it is important to clarify which interval of time or interval of weights, sex, breed, group size and feeder space allowance, feeder design, feed form, diet composition, and environmental conditions are used in each experiment.

Third, different types of pigs according to their feeding behaviour habits were identified according to the combination of the number and size of their meals (nibbler/meal eaters) with their feeding rate (slow/fast pigs). It is important to highlight that these types of pigs may exist in the same pen; therefore, there is individual variability influenced by housing conditions, individual temperament, and hierarchy within the pen. Therefore, it would be of interest to know the feeding behaviour habits of pigs with the same ADFI; this would help to evaluate the influence of the number of feeder visits, meal size, and feeding rate on feed efficiency and body composition because reducing the feeding behaviour to only the ADFI is very simplistic and does not consider those other factors. Regarding the literature reviewed, the only feeding behaviour habit found to influence feed efficiency was the time spent eating, suggesting that pigs spending less time eating have better FCR. This result could be explained by the fewer energy maintenance requirements needed. However, pigs eating faster spent less time eating, but feeding rate was not correlated with FCR. Moreover, pigs eating faster with bigger meals had higher ADFI and higher final BW, but with no differences in FCR than pigs eating slower, less, and with smaller meals; moreover, the few scientific data regarding the influence of feeding behaviour habits on carcass quality traits indicate that the former were fatter and less lean than the latter.

In conclusion, the available scientific data provide evidence that meal size and feeding rate are the two feeding behaviour habits most correlated with performance, being positively correlated with ADFI, ADG, final BW, and backfat thickness, but with no effect on feed efficiency. Therefore, more research into pigs eating the same ADFI with different feeding behaviour habits is needed to better understand the relationship between feeding behaviour habits, feed efficiency, and carcass quality traits. It is expected that the use of feeding stations and sensors in smart farming may fill the current gaps of knowledge regarding feeding behaviour and related factors; besides, other feeding behaviour parameters aside from ADFI could be considered in genetic selection programmes.

## Figures and Tables

**Figure 1 animals-12-01128-f001:**
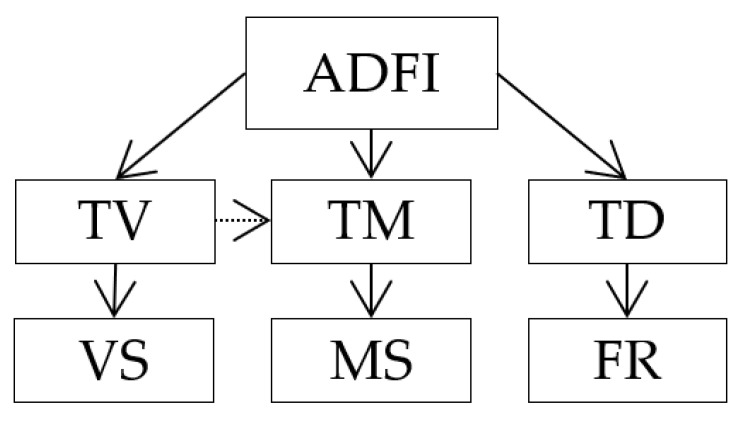
Interrelations of the feeding behaviour habits (FBHs). Average daily feed intake (ADFI), number of feeder visits per pig and day (TV), number of meals per pig and day (TM), total minutes spent eating per pig and day (TD), feed consumed per feeder visit (VS), feed consumed per meal (MS), and feed intake per minute spent eating (FR).

**Figure 6 animals-12-01128-f006:**
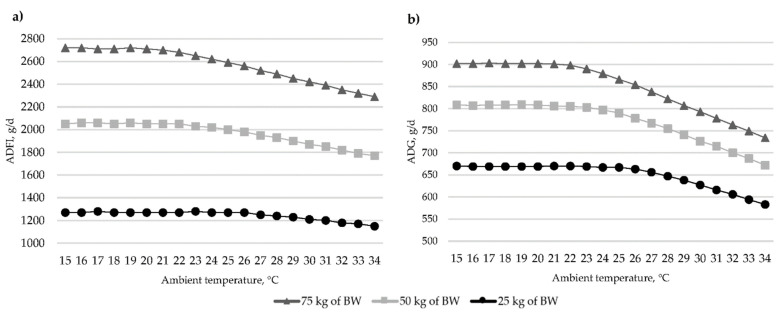
The effects of ambient temperature and pig BW on (**a**) ADFI and (**b**) ADG (Renaudeau et al. [63]).

**Table 1 animals-12-01128-t001:** Individual feeding behaviour parameters and the criteria used to compute them.

Parameter	Nomenclature	Criterion
Average daily feed intake (kg/d)	ADFI	Total feed consumed per pig and day
Feeder visits per day (n/d)	TV	Number of feeder visits per pig and day
Meals per day (n/d)	TM	Number of meals per pig and day
Time spent eating (min/d)	TD	Total minutes spent eating per pig and day
Visit size (g/feeder visit)	VS	Feed consumed per feeder visit
Meal size (g/meal)	MS	Feed consumed per meal
Feeding rate (g/min)	FR	Feed intake per minute spent eating

Average daily feed intake (ADFI), number of feeder visits per pig and day (TV), number of meals per pig and day (TM), total minutes spent eating per pig and day (TD), feed consumed per feeder visit (VS), feed consumed per meal (MS), and feed intake per minute spent eating (FR).

**Table 2 animals-12-01128-t002:** Effect of age on the feeding behaviour habits of growing-finishing pigs.

Reference	Initial and Final BW, kg	ADFI (kg of Feed/d) ^1^	TV (Feeder Visits/d) ^2^	TM (Meals/Day) ^3^	TD (Minutes Spent Eating/d) ^4^	VS (Feed Consumed/ Feeder Visit) ^5^	MS (Feed Consumed/Meal) ^6^	FR (Feed Consumed/min) ^7^
[11] **	35 to 95–100 kg	1.75 to 2.81 (increased by 60%)	From 40 to 60 kg: from 14 to 18 (increased by 28%) From 60 to 90 kg: from 18 to 16 (reduced by 11%)		From 63.7 to 49.6 (reduced by 22%)		From 278 to 621 (increased by 123%)	From 28.6 to 58.8 (increased by 106%)
[16]	27 to 82 kg	1.55 to 1.9 kg/d (increased by 23%)		From 7.25 to 6 (reduced by 17%)	From 109 to 60 (reduced by 45%)		From 220 to 320 (increased by 45%)	From 15 to 35 (increased by 133%)
[20]	40 vs. 80 kg	-	40 kg BW: 55.6 80 kg BW: 42.2 (reduced by 24%)		40 kg BW: 102 80 kg BW: 85.6 (reduced by 16%)	-	-	40 kg BW: 35.6 80 kg BW: 43.5 (increased by 22%)
[15]	30 to 100 kg	2.13 to 3.4 (increase by 60%)		From 11 to 11.3 (increased by 3%)	From 68.3 to 65.1 (reduced by 5%)		From 194 to 301 (increased by 55%)	From 31.4 to 50.2 (increased by 60%)
[5]	47 to 145 kg	Increased	Small variations		Reduced	Increased		Increased

^1^ ADFI (average daily feed intake). ^2^ TV (number of feeder visits per pig and day). ^3^ TM (number of meals per pig and day according to each paper methodology; where a meal is the successive feeder visits within two minutes [11]; the successive visits within 28.3 min intervals [16]; and the successive feeder visits within one minute [15]. Gonyou and Lou, [20] reported the number of entrances into the feeder. ^4^ TD (total minutes spent eating per pig and day). ^5^ VS (feed consumed per feeder visit). ^6^ MS (feed consumed per meal). ^7^ FR (feed intake per minute spent eating). ** Predicted values from a model.

**Table 4 animals-12-01128-t004:** The effect of feed form on the feeding behaviour habits of growing-finishing pigs.

Reference	Breed ^1^	Phase and Kg BW	Floor Space Allowance (m^2^/pig)	Feed Form and Distribution ^2^	TD (Minutes Spent Eating/d) ^3^		FR (Feed Consumed/min) ^4^	
					Pellet	Mash	Pellet	Mash
[51]	No data	25–35 kg BW	95, 110, and 125% feeder capacity	Mash vs. Pellet Dry vs. Wet–dry feeder Ad libitum	Dry: 68.9 ^b^ Wet–dry: 65.5 ^b^	Dry: 78.6 ^a^ Wet–dry: 69.7 ^b^	-	
		90–100 kg BW	80, 102.5, and 125% feeder capacity					
[53]	P × (LW × L)	8 to 26 kg BW	0.67, 0.5, and 0.4	Mash vs. Pellet Ad libitum	112.8 ^b^	175.2 ^a^	6	4
[54] **	D × (Y × L)	20 to 115 kg BW	0.8	Dry feed vs. dry feed diluted with water Twice per day	Dry: 8.6 ± 2.7 min	-	-	-
					Liquid: 3.6 ± 1.3 min			
[52]	No data (PIC)	20 to 60 kg BW	0.54	Mash-Pellet Dry vs. Wet–dry feeder Ad libitum	Dry: 81.8 ^b^ Wet–dry: 79.3 ^b^	Dry: 106.9 ^a^ Wet–dry: 71.6 ^b^	Dry: 25.9 ^b^ Wet–dry:27.2 ^b^	Dry: 19.7 ^c^ Wet–dry: 33.4 ^a^
		60 to 100 kg BW	0.76		Dry: 67.0 ^b^ Wet–dry: 65.1 ^b^	Dry: 106.5 ^a^ Wet–dry: 66.6 ^b^	Dry: 39.5 ^a^ Wet–dry: 43.4 ^a^	Dry: 25.6 ^b^ Wet–dry: 46.7 ^a^

^1^ Duroc (D), landrace (L), Large White (LW), Pietrain (P), Yorskshire (Y). ^2^ Dry or wet–dry feeder refers to different water level availability in the feeder [51,52], whereas in the study of Zoric et al. [54], pigs were fed twice per day with dry feed or with dry feed diluted with water (88.6 vs. 27.8% dry matter, dry and dry-feed diluted, respectively). ^3^ TD (total minutes spent eating per pig and day). ^4^ FR (feed intake per minute spent eating). ^a,b^ Values with different superscripts differ (*p* < 0.1). ** Mean effective time per feeding (i.e., when the first pig left the trough).

**Table 5 animals-12-01128-t005:** The effect of environmental conditions on the feeding behaviour habits of growing-finishing pigs.

Reference	Environmental Challenge	BW (kg)	Breed ^1^	Density (m^2^/pig)	Floor Type	I/GH ^2^	ADFI (kg of Feed/d) ^3^	TV (Feeder Visits/d) or TM (Meals/d) ^4^	TD (Minutes Spent Eating/d) ^5^	MS (Feed Consumed/Meal) ^6^	FR (Feed Consumed/min) ^7^
[13]	From 19 °C to 29 °C (three–four consecutive days at 19, 22, 25, 27 or 29 °C)	62 kg	P × LW	1.2 (3 pigs/pen)	Metal slatted	GH	Reduced by 24% *	Reduced by 21% **	Reduced by 28% ***	Reduced by 17%	=
[14]	13 days at 33 °C vs. at 23 °C	From 21 kg to 30 kg BW	(LW × L) × P	0.73 (5 pigs/pen)	Metal slatted	GH	Reduced by 30% **	Reduced by 30%	Reduced by 27% **	Reduced by 32% *	=
[22]	Ambient temperatures from May 2014 to April 2016	Four groups (n = 240) 4-month grow-out period	D, L and Y	0.80 (40 pigs/pen)	-	GH	-	Reduced in L pigs	4 min/d less at emergency THI level	-	-

^1^ Duroc (D), Landrace (L), Large White (LW), Pietrain (P), Yorkshire (Y). ^2^ Individual (I) or Group Housing (GH). ^3^ ADFI (average daily feed intake). ^4^ Quiniou et al. [13] and Collin et al. [14] analysed the number of meals per pig and day; according to their paper methodology; where a meal is: the successive feeder visits by the same pig within two minutes. Cross et al. [22] reported the number of feeder visits per pig and day (TV). ^5^ TD (total minutes spent eating per pig and day). ^6^ MS (feed consumed per meal: according to each paper’s methodology). ^7^ FR (feed intake per minute spent eating). * *p* < 0.05, ** *p* < 0.01, *** *p* < 0.001.

**Table 6 animals-12-01128-t006:** Correlation results between feeding behaviour habits obtained in different studies.

	TV (Feeder Visits/d) ^1^ or TM (Feeder Visits/d or Meals/d) ^2^							TD (Minutes Spent Eating/d) ^3^							VS (Feed Consumed/Visit) ^4^ or MS (Feed Consumed/Meal) ^5^						
References ^6^	1	2	3	4	5	6	7	1	2	3	4	5	6	7	1	2	3	4	5	6	7
TD (minutes spent eating/d) ^3^	0.50	−0.02	0.25	0.17	−0.06	−0.29 to 0.14	0.48														
VS (feed consumed/visit) ^4^ or MS (feed consumed /meal) ^5^	−0.76	−0.43 ***	−0.78 ***	−0.84 *	-	−0.84 * to −0.77 *	−0.84	−0.16	−0.01	−0.04	0.01	-	−0.05 to 0.30 *	−0.35							
FR (feed consumed/min) ^7^	−0.20	−0.09	0.08	−0.26 *	−0.1	−0.24 to 0.30	−0.31	−0.66	−0.76 ***	−0.59 ***	−0.79 *	−0.31 ***	−0.78 * to −0.67 *	−0.83	0.25	0.27 ***	0.14	0.34 *	-	−0.08 to 0.23	0.42

^1^ TV (number of feeder visits per pig and day). ^2^ TM (number of meals per pig and day according to each paper methodology; where a meal is: the successive feeder visits within five minutes [12]; the successive feeder visits within two minutes [11]; and the successive visits within 28.3 min intervals [16]. Young and Lawrence [24], Rauw et al. [4], Fernández et al. [26], and Garrido-Izard et al. [49] analysed the daily number of feeder visits. ^3^ TD (total minutes spent eating per pig and day). ^4^ VS (feed consumed per feeder visit). ^5^ MS (feed consumed per meal). ^6^ References: (1) [12] (Dutch Landrace, 25–35 to 100 kg BW, boars and gilts); (2) [11] (Large White and French Landrace, from 35 to 95–100 kg BW, boars and castrated males); (3) [24] (Large White × Landrace, initial weight of 32 kg BW, males and females); (4) [16] (PIC Line 26 males × Camborough females, from 27 to 82 kg BW, boars, barrows and gilts); (5) [4] (Duroc, from 38 to 130 kg BW, barrows); (6) [26] (Pietrain); and (7) [49] (Landrace, 35–50 to 107–165 kg BW, males). ^7^ FR (feed intake per minute spent eating). *, *** stand for *p* < 0.05, and *p* < 0.001.

**Table 7 animals-12-01128-t007:** Correlation results between feeding behaviour habits and average daily feed intake (ADFI).

	ADFI (kg of Feed/d) ^1^								
References ^2^	1	2	3	4	5	6	7	8	9
TV (feeder visits/d) ^3^ or TM (meals/d) ^4^	0.48	−0.06	−0.16 **	0.07	−0.28 *	−0.19 **	−0.11 to 0.01	−0.003	0.20
TD (minutes spent eating/d) ^5^	0.59	0.55 **	0.26 ***	0.51 ***	0.25 *	0.28 ***	−0.02 to 0.39 *	−0.14	0.28
VS (feed consumed/visit) ^6^ or MS (feed consumed/meal) ^7^	0.03	0.02	0.42 ***	0.40 **	0.70 *	-	0.28 * to 0.43 *	0.20 *	0.21
FR (feed consumed/min) ^8^	0.17	0.21 **	0.37 ***	0.21	0.31 *	0.26 ***	0.32 * to 0.59 *	0.51 ***	0.27

^1^ ADFI (average daily feed intake). ^2^ References: (1) [12] (Dutch Landrace, 25–35 to 100 kg BW, boars and gilts); (2) [10] (Dutch Landrace and Great Yorkshire, 25–35 to 10 kg BW, boars and gilts); (3) [11] (Large White and French Landrace, from 35 to 95–100 kg BW, boars and castrated males); (4) [24] (Large White × Landrace, initial weight of 32 kg BW, males and females); (5) [16] (PIC Line 26 males × Camborough females, from 27 to 82 kg BW, boars, barrows and gilts); (6) [4] (Duroc, from 38 to 130 kg BW, barrows); (7) [26] (Pietrain); (8) [5] (Topigs Talent × PIC, from 86 to 145 kg BW, barrows); and (9) [49] (Landrace, 35–50 to 107–165 kg BW, males). ^3^ TV (number of feeder visits per pig and day). ^4^ TM (number of meals per pig and day according to each paper methodology; where a meal is: the successive feeder visits within five minutes [12]; the successive feeder visits within two minutes [11]; and the successive visits within 28.3 min intervals [16]. Young and Lawrence [24], Rauw et al. [4], Fernández et al. [26], and Garrido-Izard et al. [49] analysed the daily number of feeder visits. ^5^ TD (total minutes spent eating per pig and day). ^6^ VS (feed consumed per feeder visit). ^7^ MS (feed consumed per meal). ^8^ FR (feed intake per minute spent eating). *, **, *** stand for *p* < 0.05, *p* < 0.01, and *p* < 0.001.

**Table 8 animals-12-01128-t008:** Correlation results between the feeding behaviour habits and growth parameters obtained in different studies.

	ADG ^1^						Final BW		FCR ^2^			
References ^3^	1	2	3	4	5	6	3	6	2	3 ^a^	6 ^a^	7 ^a^
TV (feeder visits/d) ^4^ or TM (meals/d) ^5^	0.18 **	0.01	-	−0.16 *	−0.26 * to −0.09	−0.07	−0.02	−0.11	0.00	0.14	−0.11	0.18
TD (minutes spent eating/d) ^6^	−0.06	0.17 ***	0.02	0.19 **	0.12 to 0.39 *	−0.25 *	−0.01	−0.25 *	0.15 **	−0.24 *	−0.22 *	0.33
VS (feed consumed/visit) ^7^ or MS (feed consumed/meal) ^8^	0.41 **	0.19 ***	0.38 *	-	0.28 * to 0.54 *	0.25 *	0.2 9*	0.27 **	0.02	−0.29 *	0.12	−0.08
FR (feed consumed/min) ^9^	0.50 **	0.20 ***	0.32 *	0.38 ***	0.10 to 0.43 *	0.54 ***	0.35 *	0.52 ***	−0.00	0.06	0.15	−0.16

^1^ ADG (average daily gain). ^2^ FCR (feed conversion ratio). ^3^ References: (1) [10] (Dutch Landrace and Great-Yorkshire, 25–35 to 100 kg BW, boars and gilts); (2) [11] (Large White and French Landrace, from 35 to 95–100 kg BW, boars and castrated males); (3) [16] (PIC Line 26 males × Camborough females, from 27 to 82 kg BW, boars, barrows and gilts); (4) [4] (Duroc, from 38 to 130 kg BW, barrows); (5) [26] (Pietrain); (6) [5] (Topigs Talent × PIC, from 86 to 145 kg BW, barrows); and (7) [49] (Landrace, 35–50 to 107–165 kg BW, males). ^4^ TV (number of visits per pig and day). ^5^ TM (number of meals per pig and day according to each paper methodology; where a meal is: the successive feeder visits within five minutes [10]; the successive feeder visits within two minutes [11]; and the successive visits within 28.3 min intervals [16]. Rauw et al. [4], Fernández et al. [26], Carcò et al. [5], and Garrido-Izard et al. [49] analysed the daily number of feeder visits. ^6^ TD (total minutes spent eating per pig and day). ^7^ VS (feed consumed per feeder visit). ^8^ MS (feed consumed per meal). ^9^ FR (feed intake per minute spent eating). ^a^ Gain to feed ratio. *, **, *** stand for *p* < 0.05, *p* < 0.01, and *p* < 0.0001.

**Table 9 animals-12-01128-t009:** Correlation results between feeding behaviour habits and carcass quality obtained by different studies.

	Backfat Thickness (mm)			Loin Depth (mm)	Lean Percentage (%)	
References ^1^	1	2	3	3	1	3
ADFI ^2^	0.35 **	0.36 ***	0.59 ***	0.04	−0.39 **	−0.07
TV (feeder visits/d) ^3^ or TM (meals/d) ^4^	−0.15 *	−0.07	0.06	−0.01	0.06	0.04
TD (minutes spent eating/d) ^5^	−0.05	0.08	−0.05	−0.01	−0.03	0.06
VS (feed consumed/visit) ^6^ or MS (feed consumed/meal) ^7^	0.33 **	0.16 **	0.09	0.08	−0.21 **	−0.05
FR (feed consumed/min) ^8^	0.35 **	0.13 *	0.27*	−0.028	−0.29 **	−0.06

^1^ References (1) [10] (Dutch Landrace and Great Yorkshire, 25–35 to 100 kg BW, boars and gilts); (2) [11] (Large White and French Landrace, from 35 to 95–100 kg BW, boars and castrated males); and (3) [5] (Topigs Talent × PIC, from 86 to 145 kg BW, barrows). ^2^ ADFI (average daily feed intake). ^3^ TV (number of feeder visits per pig and day). ^4^ TM (number of meals per pig and day according to each paper methodology; where a meal is: the successive feeder visits within five minutes [10]; the successive feeder visits within two minutes [11]. Carcò et al. [5] analysed the daily number of feeder visits. ^5^ TD (total minutes spent eating per pig and day). ^6^ VS (feed consumed per feeder visit). ^7^ MS (feed consumed per meal). ^8^ FR (feed intake per minute spent eating). *, **, *** stand for *p* < 0.05, *p* < 0.01, and *p* < 0.001.

## Data Availability

No new data were created or analyzed in this study. Data sharing is not applicable to this article.

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
