# Peer review of "The Feeding Behaviour Habits of Growing-Finishing Pigs and Its Effects on Growth Performance and Carcass Quality: A Review"

_animals, 2022, doi:10.3390/ani12091128_

Round 1

Reviewer 1 Report

General

the limited amount of papers on this topic has hindered the authors to make a more in depth analysis of the literature. Especially the number of papers reporting the carcass quality is very limited. The title of the paper is feeding behaviour of growing finishing pigs and its effects on growth performance and carcass quality. The number of papers to review the effects on growth performance and the number to evaluate the effects on cascass quality differ.  This leads to confusing conclusions. 

The authors state that no effect has been observed on FCR. However they mentioned that MS and Fr are positively correlated with ADFI, ADG and backfat thickness. and not with FCR, However, the reports that were used to express the effect on backfat thickness none had a negative  correlation with FCR.

The information about the feed composition and the energy content and energy density, and pellet diameter  is not provided whereas these affect the geed intake and has an effect on the change in meal seize during the fattening period. 

The authors have choosen for a monfactorial approach. It would have been valuable if they would have discussed their vision on the interaction between different factors. f.e. energy concentration and temp or feeder availability. 

In the conclusion Ishould be  a statement about the relevance of the measurements compared to ADFI.

Is it a general biological effect. Lower ADFI means less meals and/or lower FR?

The piece about the measuring equipment should be before the other data

minor remarks

line 49 Influences

Table 2  TD = minutes spent eating/d

line 142-144 could better in the paragraph diet composition

Author Response

Manuscript Number: animals-1685903 Comments to reviewers 

Firstly, we thank the reviewers for their suggestions and comments. We fully agree and think that these help us to improve the quality of the manuscript. Every change has been marked in blue.  

Reviewer 1 said: 

  • The limited amount of papers on this topic has hindered the authors to make a more in depth analysis of the literature. Especially the number of papers reporting the carcass quality is very limited. The title of the paper is feeding behaviour of growing finishing pigs and its effects on growth performance and carcass quality. The number of papers to review the effects on growth performance and the number to evaluate the effects on cascass quality differ. This leads to confusing conclusions.

It is true that the number of papers regarding the influence of the feeding behaviour of growing-finishing pigs on growth performance is much higher than the ones regarding its effects on carcass quality. Thus, when we comment the results obtained regarding carcass quality we remark that few literature exists and because of the high economic value of carcass quality it is of interest to do more research about it. However, to avoid confusion conclusions have been modified in these lines: L18-22, L32-35, L551-553.

  • The authors state that no effect has been observed on FCR. However they mentioned that MS and Fr are positively correlated with ADFI, ADG and backfat thickness. and not with FCR, However, the reports that were used to express the effect on backfat thickness none had a negative correlation with FCR.

The papers found regarding the effects of feeding behaviour habits on carcass quality traits indicate that pigs eating more and faster are fatter but with no effect on FCR. We have added this punctuation in L569-571 to avoid confusing conclusions.

  • The information about the feed composition and the energy content and energy density, and pellet diameter is not provided whereas these affect the geed intake and has an effect on the change in meal seize during the fattening period.

The diet composition is an important factor that influences the feeding behaviour habits of growing-finishing pigs. Therefore, a subsection has been written in order to indicate its importance. As the reviewer commented, the energy content is important and some cites are mentioned regarding this issue from L364 to 371. Moreover, we have searched references regarding the effect of pellet diameter on the feeding behaviour of growing-finishing with no findings. However, it has been reported that the diameter size affects the time spent eating and the preference in pre-weaned pigs but not in post-weaned pigs (Edge et al., 2005; Van den Brand et al., 2014). As we have not found literature regarding the effect of pellet diameter in the growing-finishing phase, we would appreciate if the reviewer could share with us any reference. Thank you.  

  • The authors have chosen for a non factorial approach. It would have been valuable if they would have discussed their vision on the interaction between different factors. f.e. energy concentration and temp or feeder availability.

We agree that this would have been an interesting point of view and we tried to produce a table including all these factors and their interactions, however, there is a high variability among the different papers reviewed that unable to make clear conclusions without speculating. Hence, we abandoned this idea.

  • In the conclusion it should be a statement about the relevance of the measurements compared to ADFI.

We have clarified this point. “Therefore, it would be of interest to know the feeding behaviour habits of pigs with the same ADFI; this would help to evaluate the influence of the number of feeder visits, meal size and feeding rate on feed efficiency and body composition, because reducing the feeding behaviour to only the ADFI is very simplistic and does not consider those other factors.” (L601-605)

  • Is it a general biological effect. Lower ADFI means less meals and/or lower FR?

The correlations reported by the literature between ADFI and the number of meals in group-housed growing-finishing pigs are not clear. In fact, only three out of six studies have reported a significant and negative correlation between both parameters, suggesting that pigs eating less eat in more meals per day. Regarding the correlation between ADFI and FR, four out of six studies have reported a significant and positive correlation between both parameters, suggesting that pigs eating more eat faster. This information is in table 7.

  • The piece about the measuring equipment should be before the other data.

We consider that this change would highly modify the structure and the understanding of the whole paper.

  • line 49 Influences.

It has been corrected.

  • Table 2 TD = minutes spent eating/d.

It has been corrected in all tables.

  • line 142-144 could better in the paragraph diet composition.

According to the reviewer we have replaced these lines, which now are at the end of the diet composition section (L380-384). “On the other hand, Iberian finishing pigs under extensive conditions depending on natural resources without compound feed remain actively foraging acorns and grass an average of 369 min per day, which is approximately 60 % of winter daylight hours; this kind of slow eating would be very dependent on the natural diet [60]”.

Reviewer 2 Report

The article is valuable and deals with the issues of feeding behaviour of growing/finishing pigs in great detail. I recommend this article to be printed in Animals after minor changes.

In order to make the paper more complete, I suggest the following additions and changes:

General thoughts:

- it seems that the Authors assume that the only feeding system used for growing pigs is ad libitum feeding and most of the content of the article refers to this system, but there are also others, such as the dosing system sometimes used, especially in extensive farms. What is the influence of feeding system on the behaviour of pigs? Please emphasize these issues.

- perhaps the work arrangement should be changed, i.e. at the beginning it would be best to describe what main methods of behavioural observation and registration of FBH were used by the cited researchers. It is very good that there is a section on "Automatic feeding systems used to record feeding behavior habits", but that does not yet explain what methods were used by the scientists cited in this article.

- too little attention has been paid to the influence of pigs' place in the social hierarchy, which is an extremely important factor. It is limited to one citation (46), lines 210-212. Does the position in the hierarchy make pigs nibblers, meal eaters, faster eaters, slow eaters? Moreover, individual features and temperament seem to be the most important here, while the Authors, as in most natural and animal sciences research, reduce everything to the average. A modern approach should take into account the parameters of the personality of animals. Individualization is often treated as noise or bias, while a single, individual predisposition may determine what type of feeding behaviour strategy an individual uses. Perhaps it will be possible to find and quote works with individual characteristics.

There is in the conclusion that there are different types of pigs in case of feeding behaviour, but it should be clarified in the text that it is not only about groups of pigs that behave in a specific way, but there is also individual variability, certainly depending on the position taken in hierarchy but also on the personal traits/temperament.

The above-mentioned issues are related to the palatability of the mixture (each individual may perceive it differently) and the issues of being guided by the smell. Many pig mixtures are now flavoured. Please raise these points.

Detailed Notes:

Lines: 156-157: The EU Directive 2001/88/EC is not valid!!! It is no longer in force! Date of end of validity was: 07/03/2009. It was replaced by the Directive 2008/120/EC. Please change.

Lines: 31-33: perhaps it should be noted here that it is considered only in the context of ad libitum nutrition. Besides, the sentence seems to be too long

Line 18: please change “influence” into “affect”

Author Response

Manuscript Number: animals-1685903 Comments to reviewers 

Firstly, we thank the reviewers for their suggestions and comments. We fully agree and think that these help us to improve the quality of the manuscript. Every change has been marked in blue.  

Reviewer 2 said: 

The article is valuable and deals with the issues of feeding behaviour of growing/finishing pigs in great detail. I recommend this article to be printed in Animals after minor changes.

In order to make the paper more complete, I suggest the following additions and changes:

General thoughts:

  • it seems that the Authors assume that the only feeding system used for growing pigs is ad libitum feeding and most of the content of the article refers to this system, but there are also others, such as the dosing system sometimes used, especially in extensive farms. What is the influence of feeding system on the behaviour of pigs? Please emphasize these issues.

The review is focused on the ad libitum feeding as is the most used system in intensive production. However, as other systems such as the restricted feeding system exist, we agree that it is of interest to compilate the knowledge of the feeding behaviour habits of pigs fed by restricted systems vs ad libitum system and for that we have added the results of a meta-analysis of Averós et al. [21] in which that issue is studied. On the other hand, in extensive farms pigs have restricted access to feed together with free access to grass and fodder; therefore, these pigs have more than one source to eat, which  is not comparable with restricted or ad libitum fed pigs in intensive systems. However, as this is an important topic we have included a review by Rivero et al. [55] about this topic in the new version (L361).

  • perhaps the work arrangement should be changed, i.e. at the beginning it would be best to describe what main methods of behavioural observation and registration of FBH were used by the cited researchers. It is very good that there is a section on "Automatic feeding systems used to record feeding behavior habits", but that does not yet explain what methods were used by the scientists cited in this article.

According to the reviewer, we have added a new table (Table 3) that summarizes the types of equipment used to register the feeding behaviour habits and the feeding behaviour habits measured in each study.

3)    too little attention has been paid to the influence of pigs' place in the social hierarchy, which is an extremely important factor. It is limited to one citation (46), lines 210-212. Does the position in the hierarchy make pigs nibblers, meal eaters, faster eaters, slow eaters? Moreover, individual features and temperament seem to be the most important here, while the Authors, as in most natural and animal sciences research, reduce everything to the average. A modern approach should take into account the parameters of the personality of animals. Individualization is often treated as noise or bias, while a single, individual predisposition may determine what type of feeding behaviour strategy an individual uses. Perhaps it will be possible to find and quote works with individual characteristics.

This is a very interesting appreciation. In fact, we have prepared a manuscript (non-published) in which we present a new approach to know at an individual level if pigs maintain their feeding behaviour habits during the growing-finishing period or not in order to know the individual factor. Our results show that under constant environmental conditions, most pigs maintain the feeding behaviour habits throughout the growing-finishing period, except ADFI which is the most difficult feeding behaviour habit to predict. This high maintenance together with the detection of different feeding behaviour habits within a pen (preliminar results of a non-published study that we have conducted or also observed by Hoy et al., 2012) observed indicates that the hierarchy within a pen influences the feeding behaviour habits, highlighting importance of analyzing the feeding behaviour habits at an individual level. Moreover, in our study (non-published), when environmental conditions changed from temperate to hot, most of the pigs changed their feeder visit size and the number of feeder visits indicating a sort of adaptation to hot conditions. We have included from L230-236 a paragraph to highlight the importance of the analysis of the FBH at an individual level:” Therefore, under feeder space restrictions, the hierarchy may distinctly modify FBH. Those results highlight the importance of analysing the FBH at an individual level. In fact, the authors of the present review have presented a new approach  [non-published study] to detect the maintenance of the FBH at an individual level and broadly, the results indicate that most pigs maintain the FBH throughout the growing-finishing period, except ADFI which is the most difficult FBH to predict .”

4)    There is in the conclusion that there are different types of pigs in case of feeding behaviour, but it should be clarified in the text that it is not only about groups of pigs that behave in a specific way, but there is also individual variability, certainly depending on the position taken in hierarchy but also on the personal traits/temperament.

A sentence clarifying that it is important to be aware about individual variability within a pen due to pen hierarchy or pig temperament has been included (L599-601): “It is important to highlight that those types of pigs may exist in the same pen; there-fore, there is individual variability influenced by housing conditions, individual temperament and hierarchy within the pen.”

5)    The above-mentioned issues are related to the palatability of the mixture (each individual may perceive it differently) and the issues of being guided by the smell. Many pig mixtures are now flavoured. Please raise these points.

We agree with the reviewer, and we have included these lines (L377-380): “Furthermore, the flavour and the palatability of feed may stimulate the appetite of pigs. In fact, the inclusion of flavouring additives, such as dextrose, increases the ADFI of pigs, although there are discrepancies about this fact in the literature [7].”

Detailed Notes:

6)    Lines: 156-157: The EU Directive 2001/88/EC is not valid!!! It is no longer in force! Date of end of validity was: 07/03/2009. It was replaced by the Directive 2008/120/EC. Please change. It has been changed.

7)    Lines: 31-33: perhaps it should be noted here that it is considered only in the context of ad libitum nutrition. Besides, the sentence seems to be too long. It has been changed (L32-L35): “The available scientific literature about ad libitum fed pigs suggests that pigs eating faster with bigger meals eat more, gain more weight and are fatter than pigs eating less, slower and with smaller meals. However, the feeding rate and the meal size do not influence feed efficiency..”

8)    Line 18: please change “influence” into “affect”. It has been changed.

Reviewer 3 Report

The work presents an interesting topic, since there is a lot of information in the bibliography and obtaining conclusions on this aspect is essential to continue with the scientific knowledge on animal nutrition. Next, I make a series of annotations in order to improve your manuscript.

  • Check the format of Table 2. Check that there are no loose letters.
  • Review part of the format, sometimes the title of the tables and figures are put in bold and other times not.
  • The tables and figures must be self-explanatory, any abbreviation must be explained in the same graph or table.
  • There are some figures that are not read, they must review the format. This is important, please review everyone.

  • In general, they must review the entire format of the document and unify the criteria, since it must meet the journal's standards.

  • Please do not use abbreviations in the conclusions.

  • A guide to abbreviations would be interesting and would make reading easier.

  • I think they should provide more synthetic and innovative conclusions.

  • It is a bibliographic review, they have used 79 citations on a very broad subject in a widely used species, please make sure that all the information on the subject is collected.

  • They could mark graphically how the studied parameters correlate.

Author Response

Manuscript Number: animals-1685903 Comments to reviewers 

Firstly, we thank the reviewers for their suggestions and comments. We fully agree and think that these help us to improve the quality of the manuscript. Every change has been marked in blue.  

Reviewer 3 said: 

The work presents an interesting topic, since there is a lot of information in the bibliography and obtaining conclusions on this aspect is essential to continue with the scientific knowledge on animal nutrition. Next, I make a series of annotations in order to improve your manuscript.

  • Check the format of Table 2. Check that there are no loose letters.

The table has been checked.

  • Review part of the format, sometimes the title of the tables and figures are put in bold and other times not. The tables of the paper have been checked.

  • The tables and figures must be self-explanatory, any abbreviation must be explained in the same graph or table.

 The tables and figures of the paper have been checked.

  • There are some figures that are not read, they must review the format. This is important, please review everyone. The format has been checked to meet the journal’s standards.

  • In general, they must review the entire format of the document and unify the criteria, since it must meet the journal's standards. The format has been checked to meet the journal’s standards.

  • Please do not use abbreviations in the conclusions.

The abbreviations used in the conclusions have been changed for the entire words.

  • A guide to abbreviations would be interesting and would make reading easier.

Table 1 has been created to facilitate the reading of the abbreviations.

  • I think they should provide more synthetic and innovative conclusions. Conclusions have been rewritten.

  • It is a bibliographic review, they have used 79 citations on a very broad subject in a widely used species, please make sure that all the information on the subject is collected.

We have reviewed more references than those included in the manuscript and, to our knowledge, the main references regarding this topic have been considered.

  • They could mark graphically how the studied parameters correlate.

A schema of the interrelations between the feeding behaviour habits reviewed in the present manuscript has been added in section 2 in order to clarify the understanding of the meaning of each parameter (Figure 1).

Figure 1. Interrelations of the feeding behaviour habits (FBH). Average daily feed intake (ADFI), number of feeder visits per pig and day (TV), number of meals per pig and day (TM), total minutes spent eating per pig and day (TD), feed consumed per feeder visit (VS), feed consumed per meal (MS) and feed intake per minute spent eating (FR).

Round 2

Reviewer 1 Report

The comments are answered. However, I do not agree with your comments on the place of the measuring equipment. 

Reviewer 3 Report

The authors have carried out all my suggestions